# IL-18 in Autoinflammatory Diseases: Focus on Adult Onset Still Disease and Macrophages Activation Syndrome

**DOI:** 10.3390/ijms241311125

**Published:** 2023-07-05

**Authors:** Chiara Baggio, Sara Bindoli, Irina Guidea, Andrea Doria, Francesca Oliviero, Paolo Sfriso

**Affiliations:** Rheumatology Unit, Department of Medicine, University of Padova, 35128 Padova, Italy; chiara.baggio@unipd.it (C.B.); sara.bindoli@phd.unipd.it (S.B.); irina.guidea@studenti.unipd.it (I.G.); adoria@unipd.it (A.D.); paolo.sfriso@unipd.it (P.S.)

**Keywords:** interleukin (IL)-18, autoinflammatory diseases, adult-onset Still’s disease, macrophage activation syndrome

## Abstract

Interleukin-18 (IL-18) is a potent pro-inflammatory cytokine that is involved in various innate and adaptive immune processes related to infection, inflammation, and autoimmunity. Therefore, it is described as a key mediator of autoinflammatory diseases associated with the development of macrophage activation syndrome (MAS), including systemic juvenile idiopathic arthritis and adult-onset Still’s disease. This review focuses on the role of IL-18 in inflammatory responses, placing emphasis on autoinflammatory diseases associated with chronic excess of serum IL-18, which correlate with clinical and biological signs of the disease. Therefore, it is useful for the diagnosis and monitoring of disease activity. Researchers are currently investigating IL-18’s role as a therapeutic target for the treatment of inflammatory diseases. The inhibition of IL-18 signaling through recombinant human IL-18BP (IL-18 binding protein) seems to be an effective therapeutic strategy, though further studies are necessary to clarify its importance as a therapeutic target.

## 1. Introduction

Interleukin (IL)-18 is an inflammatory cytokine that controls innate and adaptive immunity; it promotes host defense and inflammation after infection or injury [1]. It was initially described as an interferon (IFN)γ-inducing factor due to its ability to induce IFNγ production by stimulated Th1 cells. In 1995, based on the homology of its amino acid sequence with IL-1β, the cytokine’s name was changed to IL-18. Despite binding to different receptors, IL-1β and IL-18 use the same signaling pathways [2,3]. However, constitutive gene expression, synthesis, and processing are different for the two cytokines [4]. Many infectious, as well as oncologic or rheumatic diseases, have been associated with elevated levels of IL-18; indeed, it has gained increasing attention as an important mediator of autoinflammatory disorders. IL-18 stimulates a variety of inflammatory responses, enhances the proliferation and activity of T and NK cells, and shifts the helper T cell balance towards Th1 response. Extremely high total IL-18 levels have been almost exclusively observed in diseases associated with macrophage activation syndrome (MAS), including systemic juvenile idiopathic arthritis (sJIA) and adult-onset Still’s disease (AOSD), although the pathogenic and causative role of IL-18 remains unclear [2,5]. The aim of this review was to clarify both the biological properties of IL-18 and its importance in the pathogenesis of different autoinflammatory diseases, including sJIA/AOSD and MAS.

## 2. Mechanism of IL-18 Formation and Release

Hematopoietic and non-hematopoietic cells have the potential to produce interleukin IL-18 [6]. Important cell sources of IL-18 are macrophages, Kupffer cells, and dendric cells, although IL-18 is also expressed in intestinal epithelial cells, keratinocytes, astrocytes, and endothelial cells, even in the steady state and secrete functional IL-18, once activated with appropriate stimuli [2,6,7]. In contrast, human peripheral blood mononuclear cells can secrete immature IL-18 under normal conditions, suggesting that they might be cleaved extracellularly into mature IL-18 [4,7].

In humans, the IL-18 gene is located on chromosome 11 and contains seven exons with two distinct promoters, including an interferon consensus sequence binding protein (ICSBP) and a PU.1 binding site (hematopoietic-specific transcription factor). Unlike other cytokine genes, IL-18 gene has few RNA-destabilizing elements, resulting in stable cytokine expression. Transcription of IL-18 precursor can be induced via PAMPs and, ultimately, results in the activation of the NF-κB pathway. IL-18 is first synthesized as an inactive 24-kDa precursor with no signal peptide localized in the cell cytoplasm. In contrast to IL-1β, IL-18 is constitutively present and intracellulary stored as an inactive precursor in blood monocytes, macrophages, and non-immune cells from healthy subjects [3,7]. Similar to IL-1β, IL-18 is synthesized as an inactive precursor (pro IL-18) that lacks a signal peptide and needs caspase 1-mediated cleavage to become a biologically active molecule of 18 KDa [7]. Following cleavage, mature IL-18 is secreted, although over 80% of the pro IL-18 remains inside the cell [8]. Caspase-1 is activated by various inflammasomes belonging to the Nod-like receptors, AIM2-like receptors, or TRIM (tripartite motif-containing) family that contain either a CARD (caspase recruitment domain) or a PYD domain (pyrin domain). In immune cells, IL-18 is processed by NLRP3 inflammasomes, while in intestinal epithelial cells, it is processed by the NLRP6 and NLRP9b inflammasomes [2,3]. There are cases where caspase-1 processing is not required [8]. Bossaller et al. and Tsutsui et al. described a caspase-1 independent mechanism of IL-18 cleavage mediated via Fas Ligand activation that can mediate the process of IL-18 in a caspase-8-mediated fashion [9,10]. Moreover, the production and processing of IL-18 is regulated by caspase-3 [11]. In addition, other enzymes can mediate the cleavage of IL-18, such as granzyme B from cytotoxic cells, chymase from mast cells, or meprin β from intestinal and kidney epithelial cells [12,13]. Finally, IL-18 released from dying cells can be processed extracellularly in an active form by neutrophil proteases (Figure 1) [2,3].

## 3. IL-18 Signal Transduction

IL-18 mediates its effects via signaling through the IL-18 receptor (IL-18R), which belongs to the IL-1R family. IL-18R is composed of the inducible component IL-18Rα (IL-1 receptor-related protein, IL-1Rrp) and the constitutively expressed component IL-18Rβ (IL-1R accessory protein-like, IL-1RAcPL) [6]. IL-18 forms a signaling complex by binding to IL-18Rα; however, its affinity is low [3]. In cells that express the co-receptor IL-18Rβ, upon stimulation with IL-18, IL-18Rα forms a high-affinity heterodimeric complex with IL-18Rβ, which then signals [3,6]. Even if almost all cells express IL-18Rα, not all of them express IL-18Rβ. IL-18Rβ is expressed in T cells and dendritic cells, though not in mesenchymal cells [3]. In NK and T cells, stimulation with IL-12 and IFNα increases the IL-18R expression, while receptor expression can be increased through transcriptional regulation via STAT4 [14,15]. Furthermore, IL-18R is also expressed in basophils, mast cells, and CD4+ NKT cells under basal conditions [6,16]. Following the formation of the heterodimer IL-18/IL-18Rα/IL-18Rβ, the cytoplasmic Toll-IL-1 receptor (TIR) domains of the receptor complex interact with the myeloid differentiation 88 (MyD88) and initiate cell signaling via interleukin-1 receptor-associated kinase (IRAK) and TNF receptor (TNFR)-associated factor 6 (TRAF6), which, subsequently, leads to the degradation of IκB and the activation of the transcription factor NF-κB [3,6,8,17,18]. Kalina et al. reported that IL-18 involves mitogen-activated protein kinase (MAPK) cascade, as well as activation of extracellular signal-regulated kinase (ERK), c-jun N-terminal kinase (JNK), and p38 [19]. These signals induce IFNγ production and promote cell proliferation. In NK cells, IL-18 rapidly induces the activation of various molecules downstream of (phosphoinositide 3-kinase) PI3K/AKT and mammalian rapamycin target (mTOR), leading to accompanying increases in cell growth, survival, and autophagy [20]. The PI3K signal also enhances proliferation and survival in non-immune system cells, such as keratinocytes and neurons (Figure 2) [21,22].

## 4. IL-18 Mechanism of Action

IL-18 is a potent pro-inflammatory pleiotropic cytokine, which is capable of inducing an increase in cell adhesion molecules, nitric oxide (NO) synthesis, and chemokine production. In addition, IL-18 is able to induce IFNγ, granulocyte–macrophage colony-stimulating factor (GM-CSF), tumor necrosis factor (TNF), and IL-1 in immunocompetent cells to activate killing by lymphocytes, as well as upregulate the expression of chemokine receptors [3,7,8]. IL-18 mediates immune cell infiltration into tissues by increasing intercellular adhesion molecule 1 (ICAM-1) expression on myeloid cells, as well as vascular cellular adhesion molecule-1 (VCAM-1) expression on endothelial cells or synovial fibroblasts [3,23,24]. IL-18 is also essential to host defense against various infectious micro-organisms because it enhances the induction of IFNγ, NO, and reactive oxygen species (ROS). In addition, IL-18 plays a part in the clearance of viruses via the induction of CD8+ T cells, which play a central role in viral clearance [6]. IL-18 is involved in the activation and differentiation of various T cell populations. Together with IL-12, IL-18 participates in the Th1 paradigm by inducing the production of IFNγ, IL-2, GM-CSF, and IL-2Rα. IL-18 induces IFNγ either with IL-12 or IL-15 because they increase the expression of IL-18Ra. Importantly, without IL-12 or IL-15, IL-18 plays a role in Th2 diseases. IL-18 mediates the differentiation of T cells into Th2 cells through the induction of Th2-related cytokines from T cells, NK cells, and basophils/mast cells. Thus, IL-18 enhances innate immunity and both Th1- and Th2-driven immune responses [3,7]. IL-18 may negatively influence Th17 differentiation through the induction of IFNγ by Th1 cells or macrophages [3,25]. Moreover, IL-18 can be defined as an immunoregulatory cytokine as due to its prominent biological property of inducing IFNγ from Th1 cells, it also acts on non-polarized T cells, NK cells, B cells, DCs and macrophages to produce IFNγ in the presence of IL-12 via the activation of NF-κB by IL-18 and STAT-4 by IL-12 [3,6,7,8,16]. The combined stimulation with IL-18, IL-15, and IL-12 is also associated with the generation of memory-like NK cells. IL-12 and IL-18 synergistically induce IFNγ production through the reciprocal induction of their receptor expression. In addition, there is the synergistic induction of IFNγ at the molecular level. The Ifng promoter contains a consensus sequence for NF-κB, which is activated by IL-18, and the STAT4 binding site activated by IL-12, resulting in the synergistic induction of IFN-γ production [6,7]. Interestingly, the effect of IL-12 concerning IFNγ induction by T cells appears to be dependent on caspase 1; thus, they are mediated via IL-18 processing [26]. Moreover, a major immunoregulating role for IL-18 is the induction of cytotoxic actions of NK and CD8+ T cells. In particular, IL-18 directly upregulates perforin- and FasL-dependent cytotoxicity [27]. In addition, Bellora et al. described the shedding of IL-18 from the membrane of a population of M-CSF-primed macrophages, which may enable NK cells to produce IFNγ (Figure 3) [28].

## 5. Role of IL-18 Binding Protein and IL-37 in Inhibiting IL-18 Signaling

IL-18 binding protein (IL-18BP), which is present in the extracellular compartment, regulates the biological activity of IL-18. It has a high affinity for IL-18 (400 pM) and prevents IL-18 binding to IL-18R, neutralizing its biological activities in vitro and in vivo [1,2]. IL-18BP is abundant in healthy human serum, with a more than 20-fold molar excess compared to IL-18 [2,29]. Although IL-18BP is secreted, it is not defined as a soluble receptor, since it is not encoded by the same gene of IL-18 receptor and, thus, does not correspond to the extracellular ligand binding domain of the IL-18R. The amount of IL-18BP is highly regulated by the level of gene expression. IL-18BP is induced by IFNγ, suggesting that it serves as a negative feedback inhibitor of the IL-18-mediated response [1,3]. During an immune response, IL-18BP downregulates Th1 responses by binding to IL-18 and, thus, reducing the induction of IFNγ [3,6]. Other authors reported that the administration of IL-18BP substantially protects against the effect of the pathology in mouse models of experimental arthritis, colitis, endotoxin shock, and type 1 diabetes [30,31,32,33]. In patients with familial hemophagia, the failure of IFNγ to induce IL-18BP may constitute a fundamental pathogenic mechanism that exacerbates inflammation due to insufficient negative feedback [34]. Finally, IL-18BP also controls Th2 cytokine response [3].

In inflammatory diseases in which IFNγ is involved in the pathology, the concentrations of free IL-18 may be more relevant than total IL-18 in determining the severity of disease compared to IL-18 bound to IL-18BP [3,6,35]. For example, in sJIA and MAS, the level of IL-18BP is not sufficiently high enough to neutralize IL-18 [3,36]. Different studies have shown that IL-18BP therapy may be useful in situations in which high levels of IL-18 enhance the severity of disease. Indeed, clinical trials that investigate the effects of IL-18BP treatment for AOSD and NLRC4-associated MAS were previously performed (ClinicalTrials.gov Identifier: NCT02398435, NCT03113760) [37,38].

IL-37 is another endogenous factor that suppresses the action of IL-18 [6]. The tertiary structure of IL-18 has high homology with IL-37; thus, IL-18BP also binds to IL-37 [2,3,39]. Binding to IL-37 enhances the ability of IL-18BP to inhibit IFNγ induction stimulated by IL-18, though the anti-inflammatory property of IL-37 can also be affected [3,6]. Consequently, when the concentration of IL-18BP increases, IL-37 becomes less available as an anti-inflammatory cytokine. IL-37 inhibits the innate immune response by binding IL-18Rα and IL-1R8. This tripartite complex induces an anti-inflammatory response: it activates the STAT-3 signaling pathway, decreases NF-κB and AP-1 activation, and, finally, reduces IFNγ production (Figure 2) [2,6].

## 6. IL-18 in Autoinflammatory Diseases

Autoinflammatory diseases represent an expanding spectrum of inflammatory diseases characterized by self-directed tissue inflammation and the delivery of pro-inflammatory cytokines, such as IL-1, IL-18, TNF, and IFNγ derived from innate immune cells, including macrophages and neutrophils [2,40]. IL-18 plays an important role in the pathogenesis of autoinflammatory diseases, in particular in MAS or secondary hemophagocytic lymphohystiocitosis (sHLH) and sJIA/AOSD, but also in other monogenic autoinflammatory disorders that may be associated with high levels of IL-18, such as XIAP deficiency, NLRC4 gain-of-function disorders, or diseases associated with PSTPIP1 and MEVF mutations; however, in most patients with Familial Mediterranean fever (FMF), the serum levels of IL-18 are generally not significantly elevated [2,5,41].

### 6.1. The Role of IL-18 in sJIA/AOSD

Recent studies have shown that pro-inflammatory cytokines that are produced mainly by monocytes, macrophages, and neutrophils play an important role in the pathogenesis of sJIA and AOSD. Many features of sJIA and AOSD are mediated via the effects of IL-1, IL-6, M-CSF, TNF, and IL-18 [42]. IL-18 overexpression is involved in the pathogenesis of sJIA/AOSD and associated with inflammation. In AOSD patients, serum IL-18 was elevated in conjunction with the development of clinical manifestations, and interestingly, IL-18 levels even remained elevated in patients with inactive disease [43,44,45]. Therefore, IL-18 induction appears to be related to the process of systemic inflammatory disease and may cause increased circulation of IFNγ in active AOSD patients. The cellular source of serum IL-18 in AOSD has not been extensively studied, although circulating or tissue monocytes/macrophages seem the most likely candidates [42,43,44,45,46]. IL-18 plays an important role in the regulation of NK cell activity, and sJIA and AOSD are associated with NK cell dysfunction [47]. IL-18 is reported to increase the activity of NK cells; however, high levels of IL-18 may induce apoptosis in infiltrated NK cells [2,48]. Villanueva et al. reported that in patients with active sJIA an either with or without MAS, NK cell cytotoxicity was abnormal, consisting of decreased circulating NK numbers and perforin expression. In addition, impaired cytotoxic functions and/or deficiency of immunoregulatory NK cells may be relevant to the development of MAS [44]. De Jager et al. reported that the mechanism of impaired NK cell function in sJIA patients involves a defect in IL-18Rβ phosphorylation. This alteration leads to the formation of NK cells incapable of producing IFNγ in response to IL-18 [49]. Furthermore, Ohya et al. recently reported that high serum IL-18 exposure induces impaired MAPK and NF-κB phosphorylation in the NK cells of patients with sJIA [50]. Finally, the transient impairment of NK cell functions was also reported in an infant born to a mother with active AOSD and high serum IL-18 levels [2,43]. In addition, an imbalance between IL-18 and IL-18BP resulted in Th1 lymphocyte and macrophage activation [43]. The downregulation of IL-18BP expression can be mediated by miR-134. Liao et al. found that miR-134, which is upregulated via TLR3 activation, is overexpressed in the PBMC of AOSD patients, and its concentrations are also elevated in the serum of patients with active disease. Therefore, correcting IL-18/IL-18BP imbalance is a rational strategy for the treatment of AOSD [51]. IL-18 has been shown to initiate the pro-inflammatory cascade independently, without input from IFNγ. IL-18 mediates inflammation properties via direct stimulation of gene expression and synthesis of TNF from CD3+/CD4+ and NK cells. Through this mechanism, it can also indirectly stimulate CD14+ monocytes to secrete inflammatory cytokines, macrophage inflammatory protein 1α, and MCP1 [52].

Serum samples from patients with active AOSD were able to induce IFNγ production in KG-1 cells stimulated with IL-18, which was largely blocked by the anti-IL-18 antibody. Elevated levels of IFNγ were also reported in the active phase of AOSD disease, though levels of this cytokine became undetectable soon after treatment [46]. However, despite high concentrations of IL-18, IFNγ is rarely found in patients with active sJIA/AOSD. On the contrary, IFNγ may be found in the serum of patients who are affected by MAS and have increasing levels of IFNγ compared to IL-18, which may raise suspicions about the development of MAS in sJIA [3,53].

### 6.2. The Role of IL-18 in Macrophages Activated Syndrome

MAS or secondary hemophagocytic lymphohystiocitosis (sHLH) is a potentially life-threatening complication of several inflammatory disorders, including sJIA/AOSD (described in Section 6.1), systemic lupus erythematous (SLE), and infections. As MAS became more clinically recognized, an increasing frequency of occurrence in other systemic inflammatory disorders was reported [54]. MAS is characterized by excessive activation and expansion of T lymphocytes (mainly cytotoxic CD8+ T cells) and macrophages [55,56]. The activation of immune cells produces a large amount of pro-inflammatory cytokines, such as IFNγ, IL-2, M-CSF, IL-1, IL-6, IL-18, and TNF, as well as natural cytokine inhibitors, such as soluble TNF receptors and IL-1R antagonists [56,57]. MAS patients’ serum IL-18 levels are extremely elevated compared to those of other cytokines; however, the source of IL-18 during MAS is unclear [56]. Gene expression analyses of immune cells and murine tissues suggest that IL-18 is largely produced by epithelial cells, and it is also upregulated in peripheral mononuclear cells [8,54,58,59].

A contribution of IL-18 to MAS is suggested based on the role of IL-18 in the induction of IFNγ production by CD8+ T and NK cells [2,42]. However, NK cell cytolytic activity is severely impaired in MAS patients due to both NK cell lymphopenia and intrinsic NK cell functional deficiency [49]. IL-18 is not only a co-stimulus for the induction of IFNγ, but it also has direct pro-inflammatory effects by inducing TNF, IL-1, IL-8, and IL-6 [54,56,60]. High levels of serum IFNγ mediate the activation of inflammatory Th1-type CD4+ T, cytotoxic CD8+ T, and/or NK cells, as well as overproduction of IFNγ [46]. A severe increase in the levels of free IL-18 may result in T lymphocyte and macrophage activation, which escapes immunoregulatory control due to NK cell cytotoxicity; this event may create conditions that favor the development of MAS [56]. IFNγ activates macrophages that lead to the release of pro-inflammatory cytokines, including TNF, IL-1, and IL-6. In addition, lower IL-10 production might be related to the development of MAS [2]. Free IL-18 was shown to be highly elevated in the serum of MAS patients, a result that is in agreement with these findings, while blocking IL-18 receptor signaling attenuated the severity of MAS and IFNγ responses in IL-18BP-affected mice [61] (Figure 4).

Clinically, MAS may occur in 7 to 10% of patients with sJIA and 12 to 17% of patients with AOSD [43], as well as in patients who are already receiving IL-1 and IL-6 inhibitors [62,63,64]. Inoue et al. classified AOSD patients into two groups based on serum IL-6 and IL-18 levels; indeed, while sJIA/AOSD patients exhibiting higher IL-6 levels presented with prevalent articular manifestations, those with IL-18 dominant patterns had a more severe systemic disease and developed MAS. In addition, a serum IL-18 cut-off of >47,750 pg/mL was proposed to predict MAS onset [2,43,65]. Similarly, Natsumi et al. reported a significant increase in IL-18 levels in patients with MAS compared to those without MAS, and the impairment of the IL-18/NK cell axis was closely related to MAS development after sJIA/AOSD [43].

### 6.3. Role of IL-18 as Biomarker

In this context, IL-18 can be used as a diagnostic biomarker to identify MAS patients and, in turn, allow the start of a prompt treatment. High levels of IL-18 displayed high sensitivity and specificity in distinguishing MAS-associated sJIA from “pure” sJIA [66]; however, IL-18 is not able to differentiate between different forms of sHLH [67]. IL-18 levels are, therefore, increasingly used as a biomarker for sJIA and AOSD diagnosis [2,68,69,70]. High serum levels may be useful for the differentiation of Still’s disease from other inflammatory diseases [70,71,72,73]. Indeed, IL-18 levels were found to be significantly elevated in patients with sJIA compared to those with Kawasaki disease, TNF Receptor-associated periodic syndrome (TRAPS), SLE, juvenile dermatomyositis (JDM), and hematological conditions such as leukemia. Serum IL-18 concentrations are closely related to the course of sJIA/AOSD disease, and a slow decrease is associated with clinical remission. Kudela et al. defined a cut-off value of serum IL-18 levels that corresponded to 5000 pg/mL [2,74,75] to differentiate AOSD from other febrile diseases. In addition, recent reports showed that the imbalance of IL-18/IL-18BP might play an important role in the pathogenesis of sJIA/AOSD, and the balance of IL-18/IL-18BP could become a new indicator used to estimate the disease’s severity [2,76]. Furthermore, IL-18 levels are higher in MAS-associated sJIA/AOSD. Shimizu et al. reported that the serum IL-18 cut-off value for predicting MAS development was 47,750 pg/mL [2,65,68,69,77]. Mazodier et al. reported that in patients with MAS, free IL-18 concentration significantly correlated with both clinical activity and markers of Th1 lymphocyte or macrophage activation, such as elevated concentrations of IFNγ and soluble IL-2 and TNFα receptor concentrations [78].

### 6.4. Other Autoinflammatory Conditions Associated with High IL-18 Levels

MAS can occur as a complication of other autoinflammatory disorders. Recently, it was reported that NLRC4 mutations cause increased production of IL-18- and NLRC4-mediated MAS due to constitutive caspase-1 cleavage. Transcriptome analysis of a NLRC4-MAS patient showed upregulation of genes associated with apoptosis and dysregulation of genes associated with macrophage activation [79]. IL-18 blockade may be an effective cytokine-directed therapy in some forms of MAS. Canna et al. described a clinical case of refractory NLRC4-MAS in which the patient showed a sustained response to treatment with experimental IL-18 inhibition through rhIL-18BP [38]. IL-18BP is not commercially available in the United States, though it has been used in limited cases. The durability of the response to rhIL-18BP suggests the disruption of one of the amplification loops, which may involve IFNγ [38]. XIAP deficiency (X-linked inhibitor of apoptosis) is clinically characterized by sHLH, inflammatory bowel disease, and splenomegaly [2,80]. XIAP regulates different immune pathways, such as NOD1 and NOD2 signaling, Dectin1 signaling, TNF-receptor signaling, and, finally, the NLRP3 inflammasome; thus, the loss of XIAP causes abnormalities in inflammasome activity. The study by Wada et al. showed that markedly elevated IL-18 serum levels are associated with MAS in XIAP deficiency [81].

Recently, a novel hematological/autoinflammatory condition called NOCARH syndrome associated with the pathogenic variant of CDC42 (cell division control protein 42) was identified in four patients with neonatal onset cytopenia, dyshematopoiesis, autoinflammation, rashes, and HLH [82]. These patients showed strikingly increased production of IL-18, which caused them to be pre-disposed to the development of MAS through IFNγ induction [82,83].

Dominantly inherited PSTPIP1 mutations may provoke a spectrum of autoinflammatory manifestations characterized by the co-existence of pyogenic arthritis, pyoderma gangrenosum, and acne (PAPA syndrome). Stone et al. identified, in thirty-one PAPA and PAPA-like subjects, a chronic elevation of serum IL-18, without any sign of MAS; this result may suggest a link between pyrin inflammasome activation, IL-18, and autoinflammation without susceptibility to MAS; this relationship may, thus, also explain why subjects with FMF may present with increased IL-18 levels [5].

Amongst actinopathies, WDR1 disease has been recently described. It is characterized by autoinflammatory features, including recurrent stomatitis, difficult wound healing and immunodeficiency; [84] WD repeat-containing domain 1 (WDR1) is an actin regulatory protein that promotes cofilin-dependent actin filament turnover [85]. In 2015, Kim et al. observed that inflammation in WDR1 patients is driven by IL-18, and this mechanism reinforces the concept that the IL-18 is induced through aberrant actin depolymerization-driven pyrin inflammasome activation [86].

Finally, in COVID-19 infection, IL-18 was found to be a good prognostic biomarker that correlated with disease severity and mortality, thus confirming its central role in several hyperinflammatory conditions [87].

### 6.5. IL-18 as a Therapeutic Target

Other researchers have investigated IL-18 as a therapeutic target for the treatment of inflammatory diseases. Three IL-18 inhibitors were previously under evaluation, and clinical trials were conducted to verify the safety and efficacy of these IL-18BP or anti-IL-18 Ab formulations [88,89]. Two clinical trials evaluated the effect of IL-18 inhibition using IL-18BP in adult-onset Still’s disease and NLRC4-related macrophage activation syndrome (ClinicalTrials.gov Identifier: NCT02398435, NCT03113760) [37,38]. Regarding possible therapeutical approaches, IL-18BP tadekinig-alpha is a safe and effective treatment strategy for refractory AOSDs [37]; however, research experience in this field is still limited, and no further randomized trials have been proposed. Gabay et al. reported that a recombinant human recombinant IL-18BP, which is known as tadekinig alfa, appears to have a favorable safety profile and is associated with early signs of efficacy in patients with AOSD [37]. In addition, tadekinig efficacy was also reported in patients with MAS-associated sJIA [75], NLRC4 gain-of-function [38], and XIAP deficiency [89]. Canna et al. reported that IL-18 blockade provides general benefits in treating diseases marked by excessive free IL-18 and prevents their progression to MAS in susceptible patients. After a year of chronic IL-18 blockade, patients reported no signs of significant immunosuppression [38] (Figure 4). AVTX 007 (Camoteskimab, CERC 007, AEVI 007, or MEDI 2338) is an anti-IL-18 monoclonal antibody that was developed for the treatment of autoinflammatory diseases. Early-stage clinical development is ongoing (NCT04752371), and tolerability, safety, and efficacy will be evaluated in due course. APB R3 is a long-acting recombinant fusion protein composed of IL-18BP that is fused to anti-human serum albumin Fab fragment. Research is underway regarding its role in the treatment of AOSD. Finally, there is an ongoing Phase 2 trial (NCT04641442) that aims to evaluate the clinical efficacy, safety, and tolerability of MAS825, which is a bispecific IL-1/IL-18 monoclonal antibody, in patients with NLRC4-GOF. Rood et al. described a case report in which combined IL-1β/IL-18 blockade represented a novel therapeutic approach to SJIA-LD (Refractory Systemic Juvenile Idiopathic Arthritis-Associated Lung Disease), and it preliminarily appears to be safe and effective [90,91]. The immunostimulatory effects of IL-18 have been also investigated for cancer treatment [92].

## 7. Conclusions and Future Perspectives

In summary, IL-18 plays an important role in the pathophysiology of various autoinflammatory diseases, in particular diseases associated with MAS, including sJIA/AOSD [91]. However, the origin of chronic high IL-18 levels is varied and multifactorial in nature. Today, the steps needed for IL-18 activation in autoinflammatory diseases are widely unknown. IL-18 plays a pathogenic role in promoting T cell immunity, and it is closely associated with NK cell disfunction. From a clinical point of view, direct measurements of unbound, bioactive, and free forms of circulating IL-18 appear to be useful diagnostic markers of sJIA/AOSD and MAS. Despite the large quantity of evidence showing the role of IL-18 as a biomarker, future studies are necessary to clarify the role and importance of IL-18 as a therapeutic target in sJIA/AOSD, whether with or without MAS. Specific inhibition of IL-18 exerts therapeutic effects, indicating a pivotal role of IL-18 in inducing inflammation. Three IL-18 inhibitors are currently being evaluated in clinical trials; however, data on clinical efficacy are limited. Furthermore, tadekinig efficacy was reported in patients with MAS-associated sJIA/AOSD, XIAP deficiency, and NLRC4 gain-of-function. Given the high mortality rate associated with the MAS condition, we believe that further clinical trials of IL-18BP and anti-IL-18 antibodies are urgently needed. Finally, the use of drug therapies that specifically focus on IL-18 inhibition, in addition to the inhibition of IL-1β, represents a new therapeutic approach and preliminarily appears to be safe and effective [85,93].

## Figures and Tables

**Figure 1 ijms-24-11125-f001:**
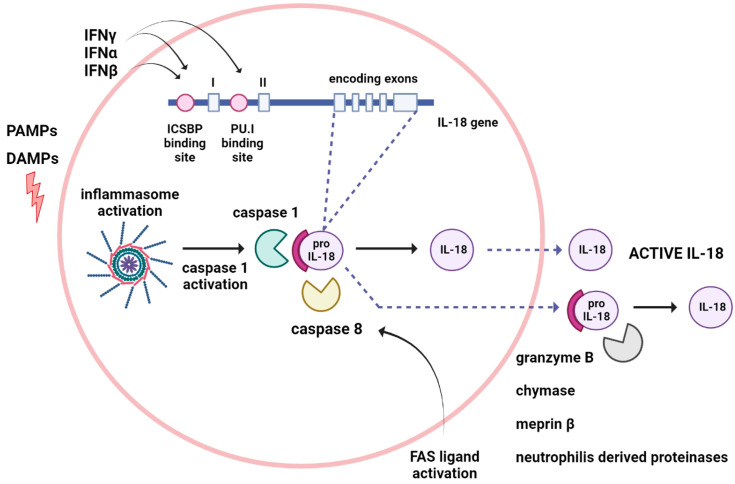
Mechanism of IL-18 formation and release. The IL-18 gene comprises seven exons, five of which are coding exons. The IL-18 promoters contain the ICSBP (interferon consensus sequence binding protein) and PU.1 (hematopoietic-specific transcription factor) binding sites. IL-18 is synthesized as an inactive precursor and needs caspase 1-mediated cleavage to become a biologically active molecule. There are cases where caspase-1 processing is not required and pro IL-18 is processed into an active molecule by granzyme B-, chymase-, meprin β- and neutrophilis-derived proteinases. Moreover, the production and processing of IL-18 is regulated by caspase-8.

**Figure 2 ijms-24-11125-f002:**
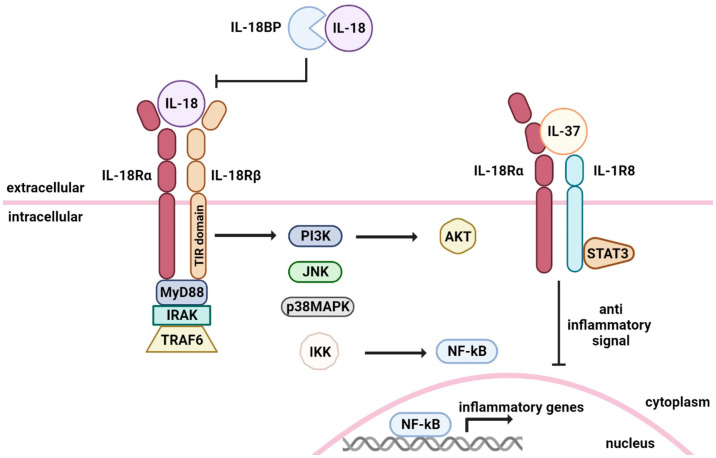
IL-18 signal transduction. When IL-18 binds to IL-18Rα (IL-1 receptor-related protein), the co-receptor IL-18Rβ is recruited to form a high-affinity complex. Following the formation of the heterodimer, IL-18 signaling activates the transcription factor NF-κB via signal transduction molecules, including MyD88 (myeloid differentiation 88), IRAK (interleukin-1 receptor-associated kinase), and TRAF6 (TNF receptor associated factor 6). IL-18BP (IL-18 binding protein) is present in the extracellular compartment and prevents IL-18 binding to its receptor. Free IL-37 binds to IL-18Rα, inducing the recruitment of IL-1R8 to form a high-affinity receptor, which, ultimately, induces an anti-inflammatory signal into the cell via STAT3.

**Figure 3 ijms-24-11125-f003:**
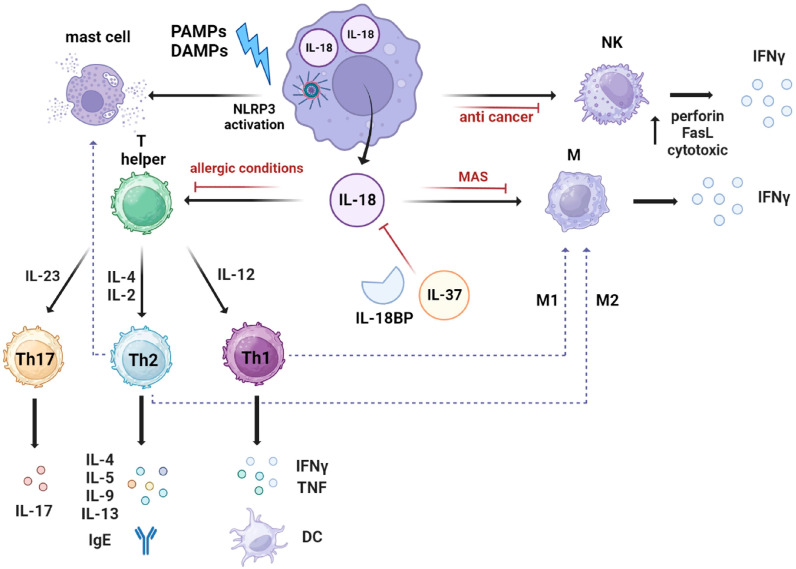
Biological functions of IL-18. IL-18 assists several immune cells, and it induces IFNγ production in macrophages, NK cells, and T cells. In addition, it participates in the Th1 and Th2 response, depending on cytokine partners. When IL-18 is combined with IL-12, it facilitates the Th1 immune response. Without IL-12 or IL-15, IL-18 promotes the differentiation of Th2 cells, which produce IL-4 and IL-13. In addition, IL-18, in combination with IL-23, promotes Th17 immune response and, finally, in response to Th2 activation, M2 macrophages and/or mast cells are activated. IL-18BP and IL-37 regulate IL-18 function; when IL-18BP (IL-18 binding protein) sequesters IL-18, the following activated cascade is inhibited. Therefore, IL-18BP can play an important role as a therapeutic target.

**Figure 4 ijms-24-11125-f004:**
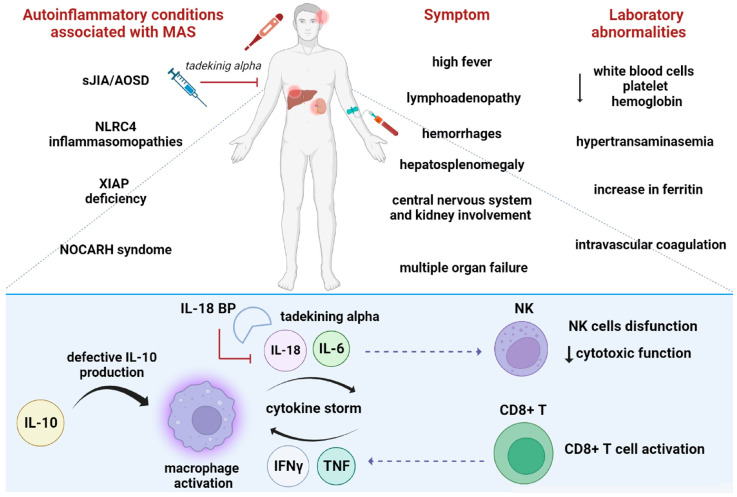
Pathogenic mechanisms of IL-18 in MAS. MAS (Macrophages Activated Syndrome) is caused by the excessive activation of T lymphocytes and macrophages. IL-10 low production, which is a regulatory cytokine that regulates IFNγ, is involved in MAS development. Overproduction of IL-18 is a hallmark of MAS; closely associated with pathogenic NK cells’ disfunction, expansion, and activation of CD8+ T cells; and, finally, overproduction of IFNγ and TNF.

## Data Availability

Not applicable.

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
