# Peer review of "IL-18 in Autoinflammatory Diseases: Focus on Adult Onset Still Disease and Macrophages Activation Syndrome"

_ijms, 2023, doi:10.3390/ijms241311125_

Round 1
Reviewer 1 Report
The Manuscript " IL-18 in autoinflammatory diseases: focus on adult onset Still disease and macrophages activation syndrome" is very well written, explaining from basics of IL-18 formation, mechanism of action, signal transduction, and functions. In addition, The elevated IL-18 effects on the immune system cells, e.g., NK cell dysfunction, proinflammatory effects on T- cells and macrophase dichotomy. Further, the author explained that altered IL-18 expressions have a role in different autoimmune diseases sJIA/AOSD, Macrophase activated syndrome, NOCARH syndrome, PAPA syndrome, WDRI disease, and COVID-19. Finally, the author discussed the ongoing clinical and pre-clinical studies targeting IL-18 for therapeutic application.
I have some suggestions to improve the article.
Figure labeling and molecular signaling or pathways depicted are too tiny and hazy; redraw them for better clarity and visual perceptiveness to better understand the audience.
The Therapeutic targeting of the IL-18 section can be improved by adding other possible methods or strategies of exploitation of IL-18 in the mentioned autoimmune disease. The author may also consider adding a new section at the end, future directions or perspectives.
Author Response
Reviewer 1
We thank the reviewer for her/his precious suggestions that will further improve our manuscript. We provide a point-by-point response to the reviewer’s comments (marked in red).
The Manuscript " IL-18 in autoinflammatory diseases: focus on adult onset Still disease and macrophages activation syndrome" is very well written, explaining from basics of IL-18 formation, mechanism of action, signal transduction, and functions. In addition, The elevated IL-18 effects on the immune system cells, e.g., NK cell dysfunction, proinflammatory effects on T- cells and macrophage dichotomy. Further, the author explained that altered IL-18 expressions have a role in different autoimmune diseases sJIA/AOSD, Macrophage activated syndrome, NOCARH syndrome, PAPA syndrome, WDRI disease, and COVID-19. Finally, the author discussed the ongoing clinical and pre-clinical studies targeting IL-18 for therapeutic application. I have some suggestions to improve the article.
Figure labeling and molecular signaling or pathways depicted are too tiny and hazy; redraw them for better clarity and visual perceptiveness to better understand the audience. We thank the reviewer for this observation. We have modified the figures as requested.
The Therapeutic targeting of the IL-18 section can be improved by adding other possible methods or strategies of exploitation of IL-18 in the mentioned autoimmune disease. The author may also consider adding a new section at the end, future directions, or perspectives. Thank you for this observation. We improved the 6.5 section “IL-18 as a therapeutic target” (lines 346 - 373) and the 7 section “conclusions and future perspectives”(lines 380 - 396).

Reviewer 2 Report
In the present review, entitled "IL-18 in autoinflammatory diseases: focus on the development of adult Still 2 disease and macrophage activation syndrome", the authors analyzed the literature on interlekin-18: its formation, mechanism of action and role in autoimmune diseases.
The work is scientifically sound, presented in a classic style and well structured.
Minor concerns
1. The Figures presented in the article are of poor quality and are difficult to read. Please correct
2. Please check the style of references [68-73] and [81]
At your request, I send comments on specific questions asked regarding the manuscript "IL-18 in autoinflammatory diseases: focus on the development of adult Still 2 disease and macrophage activation syndrome"
The work is devoted to a review of the works on Interleukin-18
(IL-18) - a powerful pro-inflammatory cytokine that is involved in various innate and adaptive immune processes associated with infection, inflammation and autoimmunity. This review focuses on the role of IL-18 in inflammatory responses, with a focus on autoinflammatory diseases associated with chronic excess of serum IL-18, which correlate with clinical and biological signs of the disease.
This topic is quite interesting and relevant, and despite the fact that
several reviews on this topic have been published recently (all of them, for example ref [3,5,6,93] are mentioned in the manuscript), the authors managed to highlight the problem from a slightly different angle.
An undoubted advantage in this review is the discussion of preclinical and clinical studies of IL-18 for therapeutic use.
In my opinion, the current work is presented in a classic style and well
structured.
I had no questions about the style of presentation and understanding of thematerial.
The main purpose of this work is to review the current situation in the
scientific world related to IL-18. Therefore, I think the authors did a good job.
All the references are appropriate.
The presented Figures have a very low image quality, they are fuzzy and muddy. In addition, the letters in the drawings are too small. Therefore, the information presented in the Figures is unreadable/ Authors should redraw the figures
Author Response
Reviewer 2
We thank the reviewer for her/his precious suggestions that will further improve our manuscript. We provide a point-by-point response to the reviewer’s comments (marked in red).
In the present review, entitled "IL-18 in autoinflammatory diseases: focus on the development of adult Still 2 disease and macrophage activation syndrome", the authors analyzed the literature on interlekin-18: its formation, mechanism of action and role in autoimmune diseases. The work is scientifically sound, presented in a classic style and well structured.
Minor concerns
The Figures presented in the article are of poor quality and are difficult to read. Please correct. We thank the reviewer for this observation. We have modified the figures as requested.
Please check the style of references [68-73] and [81]. Thank you for this observation. We have changed the style of references.
